# Modeling Self-Rollable Elastomeric Films for Building Bioinspired Hierarchical 3D Structures

**DOI:** 10.3390/ijms23158467

**Published:** 2022-07-30

**Authors:** Lorenzo Vannozzi, Alessandro Lucantonio, Arturo Castillo, Antonio De Simone, Leonardo Ricotti

**Affiliations:** 1The BioRobotics Institute, Scuola Superiore Sant’Anna, Piazza Martiri della Libertà 33, 56127 Pisa, Italy; alessandro.lucantonio@santannapisa.it (A.L.); arturocu92@gmail.com (A.C.); antonio.desimone@santannapisa.it (A.D.S.); leonardo.ricotti@santannapisa.it (L.R.); 2Department of Excellence in Robotics & AI, Scuola Superiore Sant’Anna, Piazza Martiri della Libertà 33, 56127 Pisa, Italy

**Keywords:** microfabrication, self-rolling, polydimethylsiloxane, bioinspired materials, programmable deformation, bilayer

## Abstract

In this work, an innovative model is proposed as a design tool to predict both the inner and outer radii in rolled structures based on polydimethylsiloxane bilayers. The model represents an improvement of Timoshenko’s formula taking into account the friction arising from contacts between layers arising from rolling by more than one turn, hence broadening its application field towards materials based on elastomeric bilayers capable of large deformations. The fabricated structures were also provided with surface topographical features that would make them potentially usable in different application scenarios, including cell/tissue engineering ones. The bilayer design parameters were varied, such as the initial strain (from 20 to 60%) and the bilayer thickness (from 373 to 93 µm). The model matched experimental data on the inner and outer radii nicely, especially when a high friction condition was implemented in the model, particularly reducing the error below 2% for the outer diameter while varying the strain. The model outperformed the current literature, where self-penetration is not excluded, and a single value of the radius of spontaneous rolling is used to describe multiple rolls. A complex 3D bioinspired hierarchical elastomeric microstructure made of seven spirals arranged like a hexagon inscribed in a circumference, similar to typical biological architectures (e.g., myofibrils within a sarcolemma), was also developed. In this case also, the model effectively predicted the spirals’ features (error smaller than 18%), opening interesting application scenarios in the modeling and fabrication of bioinspired materials.

## 1. Introduction

Rolled-up structures are based on flat layers of materials coupled together that get out of the bidimensional (2D) plane, becoming tridimensional (3D) assemblies. This transformation can be achieved thanks to the mismatch of physical properties (e.g., thermal expansion coefficients) or pre-stretches applied between layers [1,2,3]. The generation of non-uniform stresses and elastic deformations along with the thickness of a single layer or coupled layers can lead to the formation of structures such as cylinders, spirals, helices, and origami-like geometries [4,5]. The roll-up of a rectangular “free” membrane is bidirectional in nature, and different approaches have been proposed in the state-of-the-art to obtain predictable rolled-up structures [6,7,8,9].

Such a paradigm has been widely investigated at the mesoscale, microscale, and nanoscale using multiple materials such as semiconductors, metals, and synthetic/natural polymers. For example, micro engines or energy storage devices have been developed using rolled membranes at small scales [10,11]. Rolled-up technology has also been proposed for biomedical applications, specifically for building bioinspired structures and drug delivery systems [12].

Furthermore, the rolling of structures allows introducing a surface topographical pattern into 3D assemblies by providing the material with specific features in its 2D state, before rolling (e.g., with photolithographic processes) [13,14]. This can be useful in cell/tissue engineering applications, in which cell fate and behavior are strongly influenced by surface topography [15,16]. The strain-induced rolling mechanism (SIRM) enables the rolling of a topographically defined elastomeric bilayer structure by pre-stretching one of its layers, thus generating a stress mismatch [17,18]. The deformation provided to one of the components leads to differential stresses along with the thickness of the bilayer and, in turn, curvature. Despite the noticeable efforts in recent years in devising novel approaches for the fabrication of rolled-up structures, their modeling has become essential to elucidate the effect of key design parameters and material effects on the final structure.

The continuum mechanics of bilayers was first studied by Timoshenko [1]. He set the problem of predicting the curvature of a bimetal strip by analyzing its deformation due to uniform heating. In such a model, the curvature of a bilayer structure is proportional to the thermal expansion mismatch between its layers and inversely proportional to the bilayer thickness.

Timoshenko’s model has been applied to many other experimental scenarios where discrepancies between layers may arise due to a variety of specific physical mechanisms, being a compact way to describe similar physical problems. Nonetheless, many researchers have refined this equation when applied to systems different from the heated bimetallic strip in terms of materials, methods, and scales. For example, discrepancies can happen due to the contemporary presence of other physical effects, such as surface tension or capillary forces [19]). Other authors focused on hydrogels, highlighting the importance of considering that Timoshenko’s formula was thought for elastic deformation, while in hydrogels, the viscoelastic behavior is predominant [20]. An adaptation of Timoshenko’s formula has been proposed to understand hydrogel rolling behavior by differential swelling [21].

Despite the mentioned efforts to improve model accuracy in different scenarios, Timoshenko’s model defines a single curvature value to describe the rolled structure. This approach does not take into account the self-contact of the bilayer upon rolling. In the case where self-contact occurs, there is the need to distinguish the radius the bilayer would assume if it had vanishing thickness (*spontaneous radius*, which is the inverse of spontaneous curvature evaluable with Timoshenko’s formula) and the geometrical features that the physical roll exhibits, namely an *inner* and an *outer radius*.

For those geometries, a modeling approach that considers both inner and outer radii beyond Timoshenko’s analysis is important when requiring precise control of the geometrical features of the final assembly. The size of the inner roll must be known precisely, as well as the layer thickness with respect to the nominal radius, to avoid inaccurate results. Several examples of architectures, such as 3D tubular and hierarchical structures found in nature in muscles, tendons, and other tissues [22], would benefit from optimized models to aid researchers in creating similar geometrical features during the fabrication of the final assembly. Hierarchical spiral structures have been previously reproduced without the control of their radii and with a non-optimized design, potentially leading to poorly predictable results [4,5].

Here, we propose a new bidimensional model to predict the inner and outer radii of elastomeric bilayer structures made from polydimethylsiloxane (PDMS), provided with specific topographical features, as previously shown by our group adopting the SIRM technique [18]. Firstly, the model was exploited to predict the inner and outer radii of the bilayered structure by varying the applied strain (20, 40, and 60%) and comparing its performance, in the presence and absence of friction. Then, the model with the friction assumption was tested to predict the inner and outer radii while varying the bilayer thickness. Finally, a complex 3D bioinspired hierarchical elastomeric microstructure made of seven spirals arranged like a hexagon inscribed in a circumference was built based on the model results.

## 2. Results and Discussion

### 2.1. General Workflow and Fabrication of Topographical Features

Figure 1a reports the process followed to fabricate polymeric rolled bilayers and the conceptual workflow used to validate the model, thus optimizing the fabrication of singular spirals. Once fabricated (Appendix A), each spiral was analyzed to determine the parameters of the model that allow estimating the inner and outer radii of the bilayer, thus to predict the dimensions of the rolled bilayer (Figure 1b).

Briefly, PDMS was cast on silicon wafers, previously fabricated using specific lithographic masks and photolithographic processes, then spin-coated to obtain the desired thickness. Then, the bilayers were built by combining a stretched PDMS top layer featuring microgrooves (width: ~10 µm, height: ~1.1 µm; Appendix A) and a semi-cured bottom layer characterized by rectangular micropillars (length: 313.2 ± 2.6 μm, width: 61.3 ± 2.1 μm, height: 23.6 ± 0.4 μm). Figure 1c reports a rolled PDMS structure, highlighting the above-mentioned topographical features, namely microgrooves in the inner part of the spiral and micropillars on the outer part.

The choice of keeping the groove-based microtopography in the inner part was due to the possible future perspective (not the purpose of this paper) of cultivating multiple cell types within a rolled scaffold, aligning them along a specific direction. At the same time, the pillars on the outer surface would allow maintaining a spacing between layers, enabling nutrient diffusion. This procedure enables 3D scaffold fabrication without the need for internal vessel-like structures [10,23,24].

### 2.2. Model of Rolled Bilayers

Our samples consist of layers of lateral width w, large compared to their thickness, and their response is independent of w. For this reason, we decided to describe the rolled bilayers through a two-dimensional model representing a portion of the bilayer of unit width.

We first observed that Archimedean spirals of the form r(θ) = a + bθ fit the experimental data regarding the midlines of the rolled bilayers very well (see Materials and Methods). In particular, assuming that the rolled structure is perfectly packed (i.e., there are no gaps between successive turns), the parameter b, which controls the radial growth per turn, equals h/2π, where h is the thickness of the bilayer. Based on these observations, we developed a bidimensional model to determine the equilibrium inner radius *r_i_* of the spiral that minimizes the bending energy for a given contour length *L*:(1)$$r_{i}=\arg\underset{a}{\min}\int_{0}^{L}{\left[k\left(\theta \left(s\right);a\right)-k_{s}\right]}^{2}ds$$
where s is the arc-length coordinate, k is the curvature of the bilayer, computed as follows:(2)kθs;a=2b2+a+bθs2a+bθs2+b232
and k_s_ is its spontaneous curvature, i.e., the curvature the roll would attain if its thickness vanished. Indeed, the spiral shape results from the finite thickness of the bilayer, which prevents assuming a cylindrical (constant curvature) configuration. The spontaneous curvature was computed using the finite bending solution based on the Neo-Hookean constitutive model [21] as a function of the ratio α between the shear moduli of the layers, the ratio β between the thickness of the top layer and the total (unstrained) thickness, the pre-stretch λ, and the total (unstrained) thickness h of the bilayer. The values of these parameters, along with that of the parameter L, were estimated from experiments. This calculation refines Timoshenko’s solution [1], accounting for the non-linear constitutive response of the layers. More details can be found in the Appendix A. The relation between the angular coordinate and the arc length needed in Equation (1) was found by integrating the following equation numerically:(3)$$\frac{d\theta }{ds}=\frac{1}{\sqrt{{\left(a+b\theta \right)}^{2}+b^{2}}}$$

Finally, the outer radius of the spiral was computed as:(4)$$r_{o}=r_{i}+b\theta \left(L\right)$$

We repeated the calculations for the values of the pre-strains applied in the experiments (20%, 40%, 60%). Energy minimization was performed numerically using the Nelder–Mead algorithm provided by the “minimize” function of the SciPy library. 

To improve the match between the theory and experiments, we propose a simplified model of the frictional interactions occurring between overlapping layers that form the Archimedean spiral during rolling. Indeed, friction plays an essential role in the resulting geometry, since it may hamper sliding between successive rolls, thus leading to a geometry that differs from the configuration of the minimal bending energy. To model the rolling process, we considered a sequence of bilayers with increasing length L_n_, n = 0, … , N, where L_0_ = L_in_ and L_N_ = L are the lengths corresponding to the formation of the first roll and the total length of the bilayer, respectively. We assumed that L_in_ = 2π/k_s_, since the system can roll following the spontaneous curvature until self-contact occurs upon completing the first turn. Then, if the friction is high enough to hamper any sliding, the inner radius of the spiral at each step of the rolling process is practically fixed to its initial value: r_i,n_ = 1/k_s_, n = 0, 1,…, N. The outer radius can then be computed from equation (4). We found that the inner and outer radii evaluated under the assumption of high friction matched well the experimental data for the pre-strains of 20%, 40%, and 60%.

### 2.3. Influence of the Applied Strain on the Geometry of the Rolled Structure

The first set of analyses was based on the investigation of the top layer strain influence on the final shape of the rolled structure. Therefore, the PDMS top layer was stretched at different strains to evaluate the outcomes by varying the strain mismatch (20%, 40%, and 60%), thus the internal stresses generated upon coupling the membranes. Figure 2a illustrates a scheme of bilayers subjected to different strains and representative examples of the spiral fitting process applied for each condition. The spiral fitting method (Appendix A) provided a robust method to verify reliable measurements of the geometrical characteristics of the rolled samples, which were fit with Archimedean spirals.

In all strain cases evaluated, the thickness of the final bilayer was calculated, finding no statistically significant differences among the three groups (Appendix A). Even the distribution between the top and bottom thicknesses resulted in being the same for the different experimental groups (Appendix A). These results confirmed that the influence of the strain only concerned the final radii of curvature. The final bilayer length was also calculated, resulting in 13.73 ± 0.39 mm, 12.86 ± 1.75 mm, 12.11 ± 0.86 mm for 20%, 40%, and 60% strain values, respectively, with respect to the initial 15 mm, demonstrating that a larger contraction occurred concurrently with a larger strain. It is noticeable that the increase of strain mismatch between the top and bottom layer drove the bilayer structure towards smaller inner radii. Consequently, the number of turns increased.

The experimentally measured inner and outer radii were compared, in Figure 2b and Table 1, with the ones predicted by our model (with friction and no friction assumptions) and the nominal radius (outcome of Equation (2)). It can be observed that the spontaneous radius derived from Equation (2) fits the inner radii values. This suggests that our experimental system is characterized by high friction between the layers and that the buildup of elastic energy in the layers that are rolled after the first turn is not sufficient to trigger interlayer slip. The inner layer radius stays close to its initial value, the one established with the first turn, which coincides with the nominal radius.

Data showed a percentage error below 10% with respect to the experimental average value in predicting the outer diameter, with a better performance in the presence of friction (absolute error lower than 2.5%). The discrepancies between the experimental values and our model were more marked for the inner radius. Indeed, the model tended to underestimate the radius. Nonetheless, the presence of friction limited the absolute error with respect to the average value experimentally evaluated below 25% (worst case at 60% of strain), while the no-friction assumption led to error values from 20.5% (20% of strain) up to 51.86% (60% of strain). As shown in Figure 2b, the relatively good agreement of our model with the experimental data relies on the presence of significant friction between successive rolls in contact. However, other variables (e.g., edge effects) could lead to a slight gap between model and experiment data despite the improvements obtained by introducing the friction assumption.

### 2.4. Influence of the Thickness Variation of the Bilayer on the Geometry of the Rolled Structure

As a second step, we evaluated the robustness of our model toward changes in the bilayer thickness. For this purpose, we fabricated PDMS layers at different spin coating speeds (500, 700, 900, 1100, 1300, and 1500 rpm), then assembled in bilayers by applying a strain of 60% to the top layer (Figure 3a). We chose to apply the strain of 60% to elucidate the performance of the model in the case with the lowest inner and outer radii, which may better fit the size requirements at a smaller scale when replicating biological tissues [25].

Firstly, the fabricated samples were characterized in terms of spiral fitting. The bilayer thickness was also analyzed, showing a decreasing trend, ranging from 372.9 ± 47.8 for 500 rpm to 93.5 ± 4.6 µm for 1500 rpm (Figure 3b). We also observed that the bottom layer became thicker than the top one as the spin speed decreased, probably because of the presence of the pillars in the bottom mask, which may favor PDMS stagnation upon spinning due to the roughness they provided (Appendix A).

The model predictions for the radii were compared with experimental data, and the results of this comparison are shown in Figure 3c. The model predictions were made under the assumption of high friction, having similar results as the spontaneous radii prediction model, as shown in Figure 2b.

The spin speed variation influenced the final radius of curvature obtained. Indeed, we noticed a reduction of the inner radii from 963 ± 93 µm (500 rpm) to 359 ± 75 µm (1500 rpm). Model results obtained assuming high friction nicely matched the experimental data, with a maximum error with respect to the average radii value of 25% (Table 2). 

In this work, the strain was varied to verify its effect as a design parameter. On the other hand, thickness is an important parameter to be considered, being the most influential variable in determining the flexural rigidity of thin film, which may influence the rolling capability of each bilayer [26]. Our results showed a significant decrease in the inner and outer radii while decreasing the PDMS bilayer thickness for a given strain, meaning that the thickness is a parameter that can be easily modified for tuning the features of the rolled structure during an a priori design of the construct, e.g., for building bioinspired structures.

### 2.5. Development of a Bioinspired Hierarchical Structure

Once the predictions on the inner and outer radii of the spirals generated from specific fabrication parameters were validated under the high friction assumption, we targeted the design of more complex bioinspired structures made of multiple rolled systems. We designed a tube-in-tubes structure, promoting a compact arrangement of spirals within a tubular matrix.

As a proof-of-concept, we fabricated a hierarchical rolled structure with seven inner rolled units, all wrapped along with two turns during the rolling of an outer one, to obtain a stable assembly arranged like a hexagon inscribed in a circumference, similar to some biological architectures (e.g., myofibrils within a sarcolemma). The hierarchical structure was defined by choosing the single spirals as the fabrication parameters, resulting in a minimum inner radius (spin coating speed: 1500 rpm, strain: 60%, shear moduli ratio: 1) and a length value to generate two turns after rolling. These requirements were defined to minimize the overall cross-sectional area of the final structure by minimizing the central void space on every inner unit. The outer diameter, which had the function of unifying all inner spirals into a compact multivessel structure, was estimated to wrap all the inner units in an ordered fashion with one turn. The number of turns as a design parameter guarantees stability against external disturbances such as compressive forces transmitted by manipulation.

The fabrication process of hierarchical rolled bilayers was adapted by modifying the last two steps, thus redesigning the stretching system and wafer supports to fit with more extended PDMS membranes (Figure 4a). Once the stretched top layer was applied to the bottom one, a laser cutter machine was employed to cut a specific geometric pattern on the bilayer PDMS surface. The knowledge of the inner roll size and the outer radius of the roll dictated the definition of such a geometrical path, as the distance between the cuts in the initially flat sheet that would produce hierarchical rolling when the membrane is released. The model allowed predicting the resulting structure in terms of the geometric characteristics of the inner rolls, their number of units, and the number of outer rolls’ wrapping turns. The spiral that best fit the data points was computed by minimizing the square distance between them in the previous steps, which adhered to the experimental results.

Firstly, we derived the expected contour length of each inner spiral based on the target inner radius (0.237 mm), the number of turns (N = 2), and the bilayer thickness (93.5 µm, for the 1500 rpm speed), calculating the contour length according to the Archimedean spiral equation, as follows:(5)$$L=\int_{0}^{4\pi }\sqrt{r^{2}+{\left(\frac{dr}{d\theta }\right)}^{2}}d\theta$$

A margin (1.5 mm) was added to this value to ensure a gap between rolls. A simulation was performed to predict the inner and outer radii for the single spirals (length different from the analysis of Section 2.4, namely 15 mm), and we compared the simulation results with the experimental ones. The remaining space (called “rolled”) was kept free from cuts for a distance calculated as the length needed to cover with one turn the hierarchical structure. Following equation (5), the contour length of the spiral based on the inner radius (1.44 mm), number of turns (N = 1), and bilayer thickness (93.5 µm) was set at 9.34 mm. Thus, the total length of the bilayer for building a hierarchical structure with seven inner rolls was:(6)$$Length=6*\left(L+margin\right)+L+Rolled$$

Finally, each inner spiral’s width, namely W, was set at 0.8 mm and spaced by 0.4 mm among each external edge. The width did affect the resulting inner and outer radii.

The result was provided as the best distribution of inner units into the bilayer film and the total bilayer length. This information made it possible to plot the pattern of cuts that had to be executed by the laser cutter to obtain the desired structure after rolling (Figure 4b). Such a configuration tends to roll from its left side towards the right side to form inner spiral units within the outer diameter.

Figure 4c shows images of a hierarchical rolled structure acquired through Scannign Electron microscopy (SEM). Once the hierarchical rolls were fabricated, the size of real samples’ radii extracted from the spiral fitting process against the model predictions was assessed to evaluate the overall agreement (Figure 4d). Table 3 shows that the average values of the inner and outer radii predictions for the single spiral units were different by a percentage of 18.56% and 1.01% than the average real inner and outer radii measured experimentally.

Unlike the results reported in Table 1, the inner radius was overestimated by the model, probably due to the partial mechanical shrinking caused by the self-rolling of several PDMS bilayers within an external one. However, a slight variation in the model results could be considered acceptable while comparing the experimental results with the biological variability of tissues and related structures. The presence of homogeneous spirals is noticeable and quantitatively demonstrated by a low standard deviation, which shows the robustness of our approach with respect to the results proposed, for example, in [4]. Interestingly, we also achieved an internal diameter of the spiral units smaller than 400 µm, outperforming the results proposed in [4] (500 µm), in which tubes were also not provided with inner/outer topographical features. We also estimated the percentage error for the prediction of the outer radius of the outer spiral, found at 10.19% (Appendix A). Such a deviation from the outer radius measurement may be the consequence of the accumulated error on the radii of its internal units since they define the resulting geometry of the outer roll.

Several aspects of the roll-up phenomena have been analyzed, for example the investigation of the roll-up direction, the influence of the bilayer thickness, and the length-to-width ratio [27,28]. An interesting study reported multi-aligned structures’ assembly by combining magnetically sensitive tubular scaffolds [29]. Such an approach may constitute an alternative strategy to most common layer-by-layer strategies to recreate the features of a single roll on a larger scale (in width), for building a multivessel-like structure similar to those found in living organisms, such as muscles, nerves, or tendons [30]. In this work, the definition of the hierarchical structure was based on the definition of several parameters, as a specific bilayer’s thickness, the shear moduli of the layers, the desired outer and inner radii for single spirals, the number of turns of the inner spirals, and the number of turns of the outer spiral. Thus, the presented model constitutes an important advancement in the state-of-the-art of the fabrication of complex, bioinspired, and hierarchical structures. It accounts for features not considered by previous models to provide more realistic estimations of the resulting geometry of self-rollable elastomeric bilayers, such as the overlapping of layers during rolling and the friction interactions. Our model is bidimensional and describes situations in which the aspect ratio width/thickness is large enough (>10 in experiments) to have a behavior independent of width. This approach offers exciting opportunities for advancing the design of self-rollable structures by playing with the interface mechanics, a field that is considered a hot topic at present [31]. One of the most important novelties introduced by our work is the ability of our model to predict the rolled bilayers in terms of both inner and outer radii, instead of a single spontaneous radius, as has been typically reported in the literature [32]. Furthermore, we showed that an a priori design of the hierarchical structure is possible, leading to a well-defined architecture with uniform spiral geometries. To the best of our knowledge, the geometric considerations used to design from 2D a hierarchical tubes-in-tube structure have never been reported.

The versatile model developed in this study may allow the fabrication of anisotropic and hollow structures with tubular units having different sizes. The application of the proposed model for the fabrication of bilayers provided with a specific surface topography demonstrates the possibility of creating complex and optimized rolled architectures, also suitable for cell and tissue engineering. These matrices, indeed, may better resemble the anisotropic conformation of biological structures than scaffolds produced by other techniques [33].

## 3. Materials and Methods

### 3.1. Fabrication of Rolled Structures

The photolithographic masks used for patterning grooves and pillars were prepared as described in [18]. All the samples were fabricated with polydimethylsiloxane (PDMS, Dow Corning’s Sylgard 184) in a 15:1 monomer/curing agent ratio. After mixing, the PDMS was degassed in a vacuum chamber for 30 min before its use.

Silicon wafers (400 μm thick, p-type, boron-doped, <100>, Silicon Materials, Kaufering, Germany) of 3 inches were used to fabricate the patterned wafers with grooves and pillars, following the same methods proposed in [18].

The pillars’ wafer was divided into smaller square pieces of approximately 20 × 10 mm^2^, later used to cast the bottom PDMS layer, while the grooves’ wafer was used as is. Spin coating (SPS-EUROPE, SPIN150) was used to control the PDMS layer thickness. Before the PDMS casting, we covered the wafers with a sacrificial layer of polyvinyl alcohol (PVA, Mw = 23,000 g/mol, 98% hydrolyzed, Sigma-Aldrich) at 10% *w/v* in deionized water. Approximately 3 mL of PVA was poured on the wafers and spin-coated for 1 min at 2000 rpm; then, they were put into a hot plate at 80 °C for 1 min to vaporize the excess of water.

For spin coating, we poured approximately 3 g of PDMS in the center of the grooved wafer and approximately 1 g for the pillar wafer. They were spun at different velocities (from 500 to 1500 rpm) to obtain different thicknesses. Top wafers were then subjected to a baking process at 150 °C for 15 min. The top layers were then cut to a 10 mm width, peeled out of the wafer, and stretched at different strains using a custom linear rail platform. An optical microscope was used to verify the strain applied to the membrane after stretching by quantifying the variation of the distance between the grooves. Once stretched, they were fixed on 15 mm-long frames to hold the strains for the whole process. On the other hand, the bottom layers were subjected to a baking process at 80 °C for 195 s to reach a semi-cured state. Then, they were manually bonded with the framed top layers, and the bilayer was subjected to a post-baking process at 80 °C for 30 min. 

Then, the bottom wafer was carefully detached from the PDMS bilayer. Finally, one of the extremities of the bilayer was cut with a scalpel to start the rolling process. Once the rolls were formed, the remaining extreme was also cut to leave the rolled structure unconstrained for further analyses.

### 3.2. Fabrication of Hierarchical Rolled Structures

All the samples were fabricated with PDMS in a 15:1 monomer/curing agent ratio and prepared as reported above. Silicon wafers of 3 inches were used to fabricate the patterned wafers with grooves and pillars, following the same methods proposed in [9]. PVA was applied before the PDMS deposition. 

For spin coating, we poured approximately 3 g of PDMS in the center of the grooves’ and pillars’ wafers, and they were spun at 1500 rpm. Top wafers were then subjected to a baking process at 150 °C for 15 min. The top layers were then cut to a 10 mm width, peeled out of the wafer, and stretched at 60% using a customized linear rail platform to fit the 3-inch bottom wafer (Appendix A). The bottom layers were subjected to a baking process at 80 °C for 195 s to reach a semi-cured state. Then, they were manually bonded with the framed top layers, and the bilayer was subjected to a post-baking process at 80 °C for 30 min.

A laser cutting machine was used to cut the three edges of the four the inner spirals on the PDMS bilayer formed onto the PVA-covered silicon wafer, according to Figure 4b. Afterward, a few drops of deionized water were poured on the inner windows of the overall bilayers to let the inner spiral units roll. As the last step, the remaining edge of the bilayer was cut using a scalpel.

### 3.3. Morphological Characterization

A KLA Tencor profilometer was used to evaluate the thickness of the samples. Measurements at four different positions were taken for each sample to extract its average thickness value. An optical microscope (Hirox, KH-7700 Digital 3D video microscope) was also used to observe the edge of the films to evaluate their thickness.

The PDMS bilayers were analyzed by optical microscopy and scanning electron microscopy (SEM, EVO MA15 Zeiss instrument). Before using the SEM microscope, samples were coated with an ultrathin film of gold using a low-vacuum sputter coater, with a current of 20 mA and a sputtering time of 60 s.

### 3.4. Fitting of Experimental Data on Rolled Shapes

Rolled samples were placed on the optical microscope perpendicular to their rolling direction to obtain an image of their rolling profile. Those images were used for the spiral fitting process. Images were processed in MATLAB to collect a large set of points that followed the midline of the bilayer. An algorithm compared the set of points with many ideal Archimedean spiral configurations. The combination of parameters that produced the spiral minimizing the squared error between the experiment and theory was plotted and inspected visually. A set of inner and outer radii from the best-fit spirals from each sample were extracted and used to represent our experimental samples in the comparison with our model.

## 4. Conclusions

This work reported a novel model to predict both inner and outer radii in rolled structures based on a polydimethylsiloxane bilayer. The fabricated structures were provided with a surface topography, making them potentially usable in different application scenarios, including cell and tissue engineering ones. The model proved its reliability in predicting the inner and outer radii of the rolled structures when key design parameters such as the initial strain and the bilayer thickness were varied. The model better fit experimental data when a high friction assumption was used. We also demonstrated the possibility to develop complex 3D bioinspired hierarchical elastomeric microstructures made of several spirals arranged like a hexagon inscribed in a circumference, similar to typical biological architectures (e.g., myofibrils within a sarcolemma), opening interesting application scenarios in the fabrication of bioinspired materials.

## Figures and Tables

**Figure 1 ijms-23-08467-f001:**
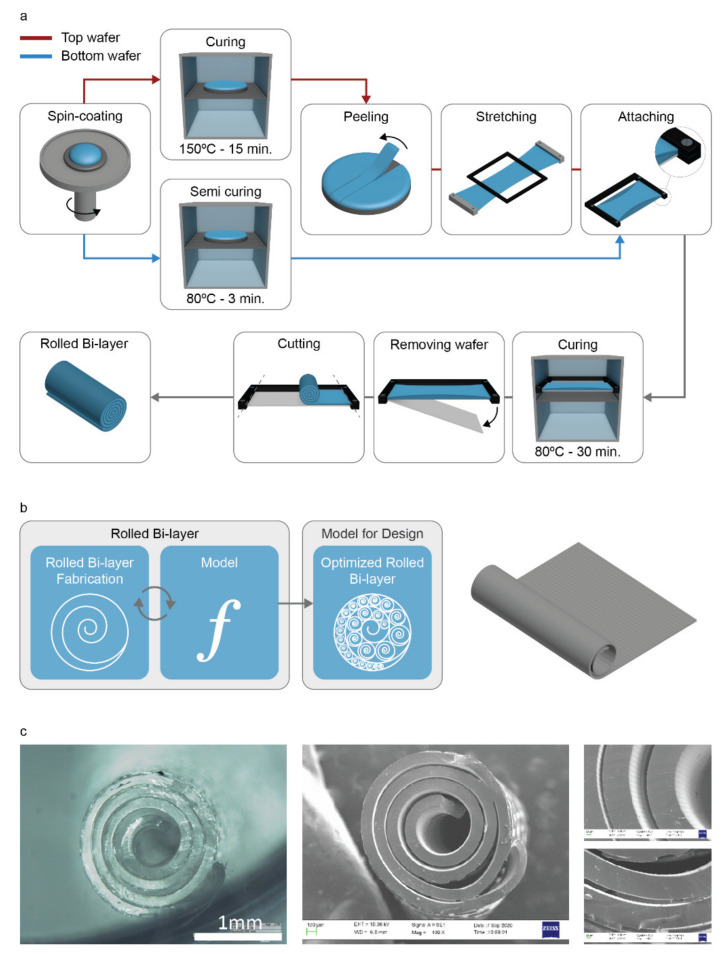
(**a**) Schematic of the fabrication process (rolled bilayers). (**b**) Depiction of the workflow followed: the bilayers’ fabrication allows validating the model. (**c**) Optical and SEM images of a rolled bilayer, fabricated using a spin coating speed of 1500 rpm, showing inner and outer surfaces provided with microgrooves and pillars, respectively.

**Figure 2 ijms-23-08467-f002:**
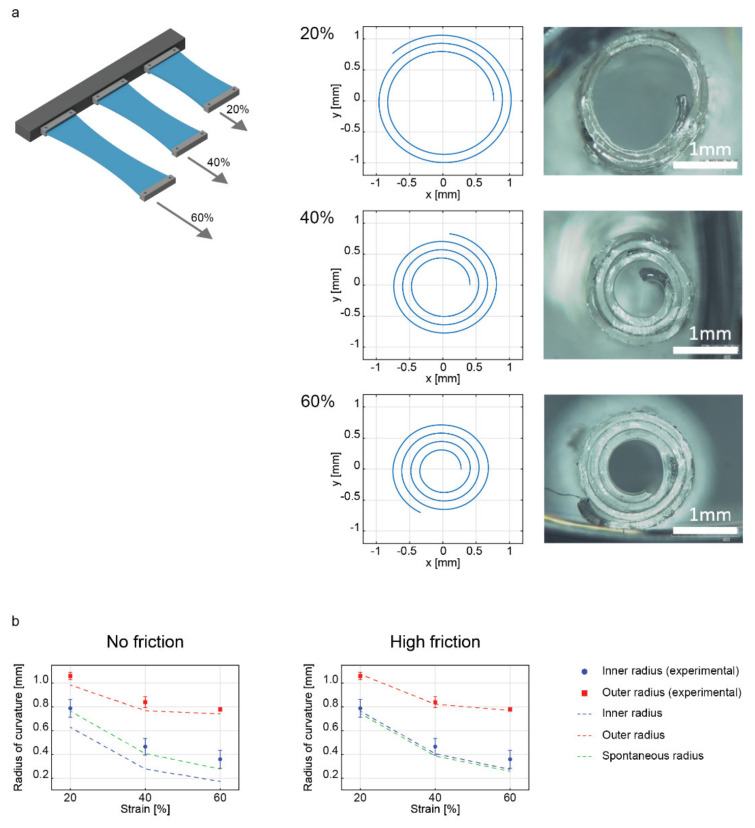
(**a**) Scheme of different strain values applied to top layers; simulation of rolled bilayers at 20%, 40%, and 60% strain based on model prediction, and microscope images of rolled bilayers fabricated using a strain of 20%, 40%, and 60%. (**b**) Plots compare experimental data and model predictions for rolled bilayers at different strain values under the assumptions of no friction (left) and high friction (right).

**Figure 3 ijms-23-08467-f003:**
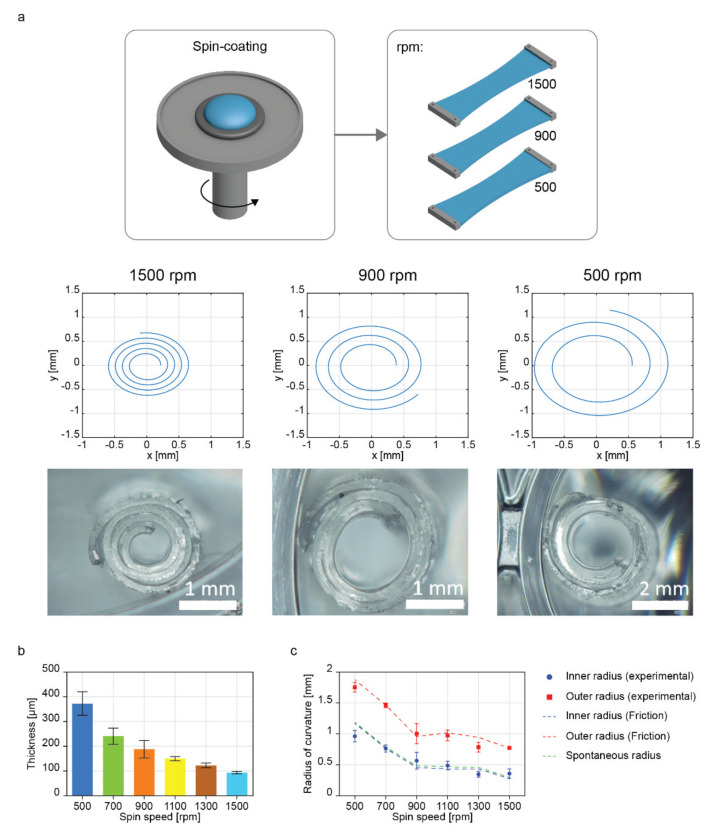
Bilayer fabrication and model validation at different thickness values. (**a**) Schematic representing the fabricated films having different thickness values by varying the spin coating speed (from 500 to 1500), representative images of model-derived spirals, and microscope images of bilayers fabricated at different spin coating velocities. (**b**) Analysis of the bilayer thickness at different spin coating speeds (from 500 to 1500 rpm). (**c**) Comparative plot between experimental data and model predictions for rolled bilayers at different thickness values under the high friction assumption.

**Figure 4 ijms-23-08467-f004:**
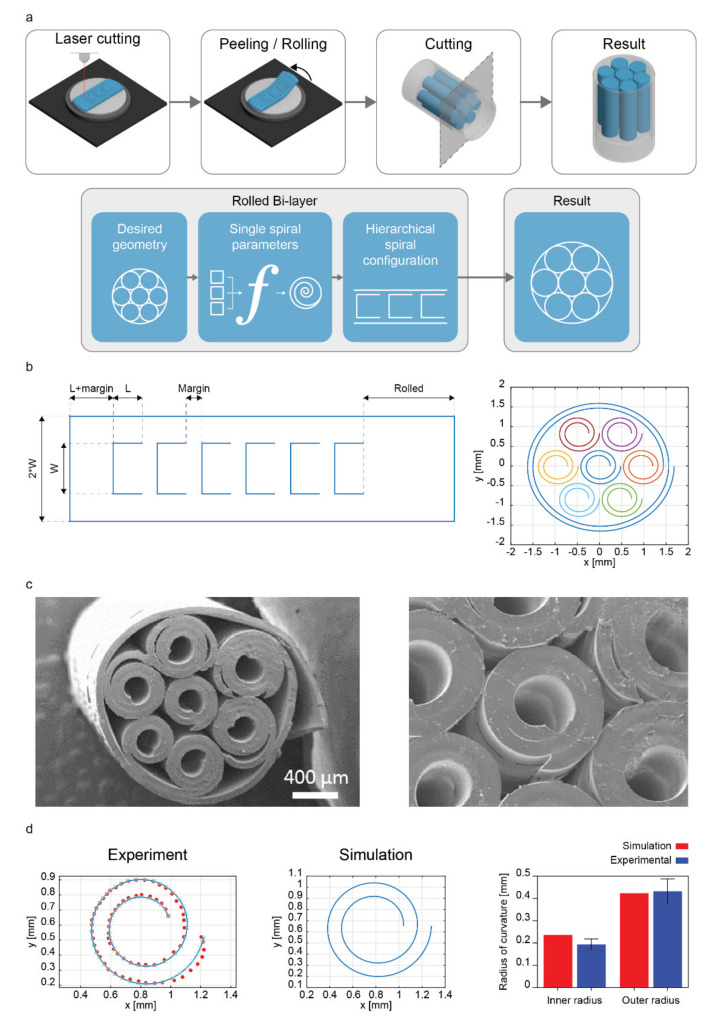
Fabrication and characterization of a bioinspired hierarchical spiral structure. (**a**) Schematic of the fabrication process of the hierarchical rolled structure. (**b**) Representation of the 2D pattern design, used to generate the hierarchical structure. (**c**) SEM images of a hierarchical rolled structure. (**d**) Spiral fitting characterization of the central inner unit of the hierarchical spiral and comparison between model (simulation) outcomes and experimental results.

**Table 1 ijms-23-08467-t001:** Comparison of the inner and outer radii, experimentally analyzed, and their prediction in terms of the spontaneous radius and in the presence/absence of the friction assumption. IR = inner radius; OR = outer radius. N = 5. The percentage difference between our model’s outcomes and the experimental results is reported in the parentheses.

Strain	Radius	Experiment (µm)	Spontaneous Radius (µm)	Model (No-Friction) (µm)	Model (Friction) (µm)
20%	IR	787.5 ± 74.8	763.7	626.2 (−20.5%)	763.4 (3.1%)
	OR	1058.2 ± 29.1		982.6 (−7.1%)	1074.0 (1.5%)
40%	IR	465.3 ± 68.7	407.3	277.5 (−40.4%)	405.9 (−12.8%)
	OR	839.3 ± 46.2		767.5 (−8.5%)	821.2 (−2.1%)
60%	IR	359.2 ± 75.7	277.5	172.9 (−51.9%)	274.1 (−23.7%)
	OR	778.4 ± 15.8		742.4 (−4.6%)	770.7 (−1.0%)

**Table 2 ijms-23-08467-t002:** Comparison of the inner and outer radii, experimentally analyzed, and their model prediction in terms of spontaneous radius, as well as in the presence of the friction assumption by varying the spin speed from 500 rpm to 1500 rpm, thus the bilayer thickness. IR = inner radius; OR = outer radius. N = 3.

Spin Speed	Radius	Experiment (µm)	Spontaneous Radius (µm)	Model (Friction) (µm)
500 rpm	IR	963.1 ± 92.9	1170.3	1173.2 (21.8%)
	OR	1754.3 ± 78.3		1875.2 (6.9%)
700 rpm	IR	764.8 ± 55.9	765.5	767.9 (0.4%)
	OR	1463.7 ± 37.3		1466.2 (0.2%)
900 rpm	IR	566.7 ± 134.4	468.6	456.1 (−19.4%)
	OR	1004.5 ± 162.1		958.3 (−4.6%)
1100 rpm	IR	489.3 ± 67.9	450.8	434.3 (−11.2%)
	OR	976.6 ± 86.6		1020.3 (4.5%)
1300 rpm	IR	348.2 ± 45.6	441.1	434.0 (24.6%)
	OR	786.2 ± 79.1		945.6 (20.3%)
1500 rpm	IR	359.2 ± 75.7	277.5	274.1 (−23.7%)
	OR	778.4 ± 15.8		770.7 (−1.0%)

**Table 3 ijms-23-08467-t003:** Comparison of the inner and outer radii, experimentally analyzed, and their prediction by the model. IR = inner radius; OR = outer radius. N = 5.

Mean Value for Inner Spirals	Experiment (µm)	Model (µm)
IR	193.3 ± 14.7	237.4 (18.56%)
OR	430.5 ± 42.0	424.1 (−1.01%)

## Data Availability

Not applicable.

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
