# Peer review of "Modeling Self-Rollable Elastomeric Films for Building Bioinspired Hierarchical 3D Structures"

_ijms, 2022, doi:10.3390/ijms23158467_

Round 1

Reviewer 1 Report

  1. Abstract: Please mention the most important results into the abstract providing specific values.
  2. Introduction: at the of the introduction, you should explain the structure of the study step-by-step. It is not clear what are you going to present.
  3. Page 2: “We propose a new model to predict the inner and outer radii of elastomeric bilayer structures made from polydimethylsiloxane (PDMS)”. Is it analytical model, semi-analytical? Please clarify at this point.
  4. Page 3: “Then, the bilayers were built by combining a stretched PDMS top layer featured by microgrooves”.

SEM pictures that show the microgrooves are required.

Also, anisotropic effects will be observed. How did you handle this? It is not clear.

  1. Page 4: “The Timoshenko's formula was modified to account for the non-linear constitutive response of the layers since the use of a large strain mismatch…”. Please elaborate more on the modification of this analytical formula. Which reference did you use? Could you explain in detail manner the modified formula?
  2. Page 5: “Through multiple iterations, the algorithm used a least-square method to compute the friction coefficient kf producing the best fit between experimental data and radii predictions” and “Once the optimal friction coefficient was found, we used this value to calculate the minimum bending inner energy radius using the previous equation, followed by calculating the outer radius using equation (5)”.

Please provide an example with at least 3 iterations using specific numbers (inputs-outputs) for reproduction purposes. This info could be implemented into the Appendix / Sup. File.

  1. How the width of the strip affects the results?

Author Response

REVIEWER 1

Open Review

English language and style

( ) Extensive editing of English language and style required
( ) Moderate English changes required
(x) English language and style are fine/minor spell check required
( ) I don't feel qualified to judge about the English language and style

Yes

Can be improved

Must be improved

Not applicable

Does the introduction provide sufficient background and include all relevant references?

( )

(x)

( )

( )

Are all the cited references relevant to the research?

(x)

( )

( )

( )

Is the research design appropriate?

( )

( )

(x)

( )

Are the methods adequately described?

( )

(x)

( )

( )

Are the results clearly presented?

(x)

( )

( )

( )

Are the conclusions supported by the results?

( )

( )

( )

( )

Comments and Suggestions for Authors

  1. Abstract: Please mention the most important results into the abstract providing specific values.

We thank the reviewer for this comment. We have modified the abstract introducing the most important results, accordingly. Here, the newest version of the abstract follows:

“In this work an innovative model is proposed as a design tool to predict both inner and outer radii in rolled structures based on polydimethylsiloxane bilayers. The model represents an improvement of Timoshenko’s formula taking into account the friction arising from contacts between layers arising from rolling by more than one turn, hence broadening its application field towards materials based on elastomeric bilayers capable of large deformations. The fabricated structures were also provided with surface topographical features that would make them potentially usable in different applica-tion scenarios, including cell/tissue engineering ones. Bilayer design parameters were varied, such as the initial strain (from 20 to 60 %) and the bilayer thickness (from 373 to 93 µm). The model matched experimental data on inner and outer radii nicely, especially when a high-friction condition was implemented in the model, particularly reducing the error below 2 % for the outer diameter while varying the strain. The model outperformed the current literature, where self-penetration is not excluded, and a single value of the radius of spontaneous rolling is used to describe multiple rolls. A complex 3D bioinspired hierarchical elastomeric microstructure made of seven spirals arranged like a hexagon inscribed in a circumference, similar to typical biological architectures (e.g., myofibrils within a sarcolemma) was also developed. Also in this case, the model effectively predicted the spirals’ features (error smaller than 18 %), opening interesting application scenarios in the modeling and fabrication of bioinspired materials.”

  1. Introduction: at the of the introduction, you should explain the structure of the study step-by-step. It is not clear what are you going to present.

We thank the reviewer for this observation, which allowed us to improve the clarity of the manuscript. As suggested, we have added a further explanation of the workflow followed in our study at the end of the Introduction, to let the reader clearly understand the content and the subsequent results, as follows:

“Firstly, the model was exploited to predict the inner and outer radii of the bilayered structure by varying the applied strain (20, 40 and 60 %) and comparing its performance, in the presence and absence of friction. Then, the model with the friction assumption was tested to predict the inner and outer radii while varying the bilayer thickness. Finally, a complex 3D bioinspired hierarchical elastomeric microstructure made of seven spirals arranged like a hexagon inscribed in a circumference was built based on model results.”

We also changed the title of section 2.2, 2.3 and 2.4 to make the content clearer to the reader, according to the workflow described.

  1. Page 2: “We propose a new model to predict the inner and outer radii of elastomeric bilayer structures made from polydimethylsiloxane (PDMS)”. Is it analytical model, semi-analytical? Please clarify at this point.

We thank the reviewer for the comment. Our model is semi-analytical, since it is based on analytical manipulations for the derivation of the equations that determine the spontaneous curvature of the bilayer (see Supplementary Materials, section: “Spontaneous curvature of the bilayer”), while numerical methods are used for the solution of the non-linear equations arising from these manipulations and for energy minimization.

  1. Page 3: “Then, the bilayers were built by combining a stretched PDMS top layer featured by microgrooves”.

SEM pictures that show the microgrooves are required.

Also, anisotropic effects will be observed. How did you handle this? It is not clear.

We thank the reviewer for the observation. We have added an image to highlight the presence of microgrooves on the top layer, derived from the photolithographic procedure used to build parallel channels.

In the Supplementary Material, the Figure S2 has been added to show the topography of the top layer, as follows:

Figure S2: SEM image of the topographic features built on the top layer. Scale bar is 30 µm.

The anisotropicity refers to the behavior of cells if they were seeded on top of the PDMS layer. In fact, for the theory of contact guidance, cells can feel the substrate to which they adhere, and their behavior (e.g., orientation) can be controlled by playing with the topography [R1]. In this sense, parallel channels are known to induce cell alignment, as demonstrated in our previous manuscript on the SIRM technique [R2]. Anisotropicity induced on the topographical surface, or in the structure, can help resemble biological structures as muscles, nerves or tendons [R3].

[R1] - Tranquillo, Robert T. "Self-organization of tissue-equivalents: the nature and role of contact guidance." Biochemical Society Symposium. Vol. 65. 1999.

[R2] - Vannozzi, Lorenzo, et al. "Self-assembly of polydimethylsiloxane structures from 2D to 3D for bio-hybrid actuation." Bioinspiration & biomimetics 10.5 (2015): 056001.

[R3] - Liu, Zengqian, Zhefeng Zhang, and Robert O. Ritchie. "Structural orientation and anisotropy in biological materials: functional designs and mechanics." Advanced Functional Materials 30.10 (2020): 1908121.

  1. Page 4: “The Timoshenko's formula was modified to account for the non-linear constitutive response of the layers since the use of a large strain mismatch…”. Please elaborate more on the modification of this analytical formula. Which reference did you use? Could you explain in detail manner the modified formula?

We thank the reviewer for this observation. We have adapted the model found in Ref. 21 for the derivation of the governing equations that allow computing (numerically) the spontaneous curvature in our setting, where the original Timoshenko’s formula is not accurate due to the large pre-stretches involved. We have added details about the calculation of the spontaneous curvature of the bilayer to the Supplementary Materials (section: “Spontaneous curvature of the bilayer”).

  1. Page 5: “Through multiple iterations, the algorithm used a least-square method to compute the friction coefficient kf producing the best fit between experimental data and radii predictions” and “Once the optimal friction coefficient was found, we used this value to calculate the minimum bending inner energy radius using the previous equation, followed by calculating the outer radius using equation (5)”.

Please provide an example with at least 3 iterations using specific numbers (inputs-outputs) for reproduction purposes. This info could be implemented into the Appendix / Sup. File.

We thank the reviewer for his/her comment. We have simplified the modeling of friction, so that a numerical optimization procedure to find the value of the friction coefficient is no longer needed. At this scope, Section 2.2 (Model description) has been modified accordingly, as follows:

“Our samples consist of layers of lateral width w, large compared to their thick-ness, and their response is independent of w. For this reason, we decided to describe the rolled bilayers through a two-dimensional model representing a portion of the bi-layer of unit width.

We first observed that Archimedean spirals of the form r(θ) = a + bθ fit experi-mental data regarding the midlines of the rolled bilayers very well (see Materials and Methods). In particular, assuming that the rolled structure is perfectly packed (i.e., there are no gaps between successive turns), the parameter b, which controls the radial growth per turn, equals h/2π, where h is the thickness of the bilayer. Based on these observations, we developed a bidimensional model to determine the equilibrium inner radius ri of the spiral that minimizes the bending energy for a given contour length L:

                      (1)

where s is the arc-length coordinate, k is the curvature of the bilayer computed as follows:

and ks is its spontaneous curvature, i.e. the curvature the roll would attain if its thickness vanished. Indeed, the spiral shape results from the finite thickness of the bilayer, which prevents from assuming a cylindrical (constant curvature) configuration. The spontaneous curvature was computed using the finite bending solution based on the Neo-Hookean constitutive model [21] as a function of the ratio α between the shear moduli of the layers, the ratio β between the thickness of the top layer and the total (unstrained) thickness, the pre-stretch λ and the total (unstrained) thickness h of the bilayer. The values of these parameters, along with that of the parameter L, were estimated from experiments. This calculation refines Timoshenko’s solution [1], accounting for the non-linear constitutive response of the layers. More details can be found in the Supplementary Materials. The relation between the angular coordinate and the arc length needed in eq. (1) was found by integrating the following equation numerically:

Finally, the outer radius of the spiral was computed as:

                     (4)

We repeated the calculations for the values of the pre-strains applied in experi-ments (20%, 40%, 60%). Energy minimization was performed numerically using the Nelder-Mead algorithm provided by the “minimize” function of the SciPy library.

To improve the match between the theory and experiments, we propose a simpli-fied model of the frictional interactions occurring between overlapping layers that form the Archimedean spiral during rolling. Indeed, friction plays an essential role in the resulting geometry, since it may hamper sliding between successive rolls, thus leading to a geometry that differs from the configuration of minimal bending energy. To model the rolling process, we considered a sequence of bilayers with increasing length Ln, n = 0, . . . ,N, where L0 = Lin and LN = L are the lengths corresponding to the formation of the first roll and the total length of the bilayer, respectively. We assume that Lin = 2π/ks, since the system can roll following the spontaneous curvature until self-contact occurs upon completing the first turn. Then, if the friction is high enough to hamper any sliding, the inner radius of the spiral at each step of the rolling process is practically fixed to its initial value: ri,n = 1/ks, n = 0, 1,…,N. The outer radius can then be computed from equation eq. (4). We found that the inner and outer radii evaluated under the assumption of high friction matched well the experimental data for the pre-strains 20%, 40% and 60%.

  1. How the width of the strip affects the results?

We thank the reviewer for the observation. Our model is two-dimensional, and it describes a strip of unit width. The model intends to describe strips of sufficient width such that the behavior of the rolls is well captured by plane strain conditions, hence independent of the lateral size. We have clarified this point in the manuscript. In our experiments, the aspect ratio is approximately 1:10. For strips of a width comparable with the thickness, it is expected that the observed behavior could become width-dependent, but these geometries appear less interesting, and they have not been explored experimentally.

Accordingly, we have added the term “bidimensional” the Introduction, as follows:

Here, we propose a new bidimensional model to predict the inner and outer radii of elastomeric bilayer structures made from polydimethylsiloxane (PDMS),…”

In section 2.2. “Model of rolled bilayers”, as follows:

“Our samples consist of layers of lateral width w, large compared to their thick-ness, and their response is independent of w. For this reason, we decided to describe the rolled bilayers through a two-dimensional model representing a portion of the bi-layer of unit width. …”

And in a statement in the Discussion section, as follows:

“Our model is bidimensional, and describes situations in which the aspect ratio width/thickness is large enough (>10 in experiments) to have a behavior independent of width.”  

Reviewer 2 Report

A most interesting paper, which I judge is rather long for the content, perhaps the authors could consider including some content in supplementary materials.

Clearly time is an important quantity in the process described, but little is said about time dependent processes. I suspect hidden in various parts of the ms, critical statements are made and it would be helpful to the reader if there was a paragraph identifying the import factors and why time dependence can be neglected.  For example, bottom of page 5 "drove the bilayer structure  towaards the smaller inner radii". On this matter there is no information on the time scale, the nature of the response, ie is there a fast component and a slow component?  

The elastic propeties of the PDMS networks will critical depend on the level of cross-linking of the PDMS> The authors do not report any characterisatisation of the network. The authors simply assume reproducibility but present no evidence of this.

Section 4 seems in conflict with the scheme in Figure 1.   Section 4 introduces a "baking" process  which presumably introduces a light cross linking to facilitate the elastic extension and holding in the frames. Section 4 has the cutting process as the first step after baking (which does not appear in Figure1) but as the last step before rolling in Figure 1.

The title of the ms refers to self-foldable but in the test only rolling is described. This is confusing as is the statement in the abstract " radius of spontaneous folding". Although the term "folding" has a number of meanings, the most common  of which is "paper folding", where the fold is localised and the folding angle well defined. The introduction has much to say about rolling but nothing about folding . Please alter the title.

Author Response

REVIEWER 2

Open Review

English language and style

( ) Extensive editing of English language and style required
(x) Moderate English changes required
( ) English language and style are fine/minor spell check required
( ) I don't feel qualified to judge about the English language and style

Yes

Can be improved

Must be improved

Not applicable

Does the introduction provide sufficient background and include all relevant references?

(x)

( )

( )

( )

Are all the cited references relevant to the research?

(x)

( )

( )

( )

Is the research design appropriate?

( )

(x)

( )

( )

Are the methods adequately described?

( )

(x)

( )

( )

Are the results clearly presented?

(x)

( )

( )

( )

Are the conclusions supported by the results?

(x)

( )

( )

( )

Comments and Suggestions for Authors

A most interesting paper, which I judge is rather long for the content, perhaps the authors could consider including some content in supplementary materials.

We thank the reviewer for appreciating our work. We have shifted some parts in Supplementary Materials to make the manuscript more concise and attractive to the reader. In the mean time, a careful revision of the manuscript allowed us to remove redundant and unnecessary part, as the previous section 4.5, shortening the lenght of at least one page.

Clearly time is an important quantity in the process described, but little is said about time dependent processes. I suspect hidden in various parts of the ms, critical statements are made and it would be helpful to the reader if there was a paragraph identifying the import factors and why time dependence can be neglected.  For example, bottom of page 5 "drove the bilayer structure  towaards the smaller inner radii". On this matter there is no information on the time scale, the nature of the response, ie is there a fast component and a slow component?  

We thank the reviewer for his/her comment, which allows us to clarify this point. Rolling is a time-dependent process, where dynamic friction, and possibly inertia and rate-dependent material processes could play a role in determining the final observed configuration. However, we have found that a static model based on the minimization of the elastic energy and the assumption of very high friction, which implies that no sliding occurs after the formation of the first turn, allows to accurately describe the equilibrium shapes after rolling, without involving the details of the transient processes that lead to such shapes.

The elastic properties of the PDMS networks will critical depend on the level of cross-linking of the PDMS. The authors do not report any characterisatisation of the network. The authors simply assume reproducibility but present no evidence of this.

We thank the reviewer for this observation. We assumed reproducibility of the PDMS crosslinking (Sylgard 184) by following well-consolidated protocols in the literature [R1], and by the past experience our group acquired in managing such a material. Indeed, we already characterized the mechanical properties of PDMS depending on the monomer/curing agent ratio that varied from 2.55 ± 0.09 MPa to 0.57 ± 0.05 MPa, from 5:1 to 20:1 respectively [R2]. Precisely, the ratio 15:1, the same we used in this manuscript, corresponded to 1.01 ± 0.07 MPa, a value similar to the ones reported in [R3].

Apart from measuring the Young’s modulus directly, we checked indirectly the level of crosslinking by measuring the thickness due to the strain variation, which resulted featured by a low standard deviation from Figure S4 (see the top layers, from 20 to 60 %), and from Figure S5 (see the top layers, only 60 %).

Figure S5: Thickness of the top (_top) and bottom (_bot) layers for the different applied strain values. The presence of pillars (23.6 ± 0.4 µm) has not been considered in this graph. Differences between the experimental groups were not statistically significant.

Figure S6: Thickness of the top (_top) and bottom (_bot) layers for different spin speed values, applying a strain mismatch of 60% at the top layer. The presence of pillars (23.6 ± 0.4 µm) has not been considered in this graph.

[R1] - Johnston, I. D., et al. "Mechanical characterization of bulk Sylgard 184 for microfluidics and microengineering." Journal of Micromechanics and Microengineering 24.3 (2014): 035017.

[R2] - Vannozzi, Lorenzo, et al. "Self-assembly of polydimethylsiloxane structures from 2D to 3D for bio-hybrid actuation." Bioinspiration & biomimetics 10.5 (2015): 056001.

[R3] – Enhancement of the surface free energy of PDMS for reversible and leakage-free bonding of PDMS–PS microfluidic cell-culture systems

Section 4 seems in conflict with the scheme in Figure 1.   Section 4 introduces a "baking" process  which presumably introduces a light cross linking to facilitate the elastic extension and holding in the frames. Section 4 has the cutting process as the first step after baking (which does not appear in Figure1) but as the last step before rolling in Figure 1.

We thank the reviewer for his/her observation. For building a hierarchical structure, the fabrication procedure adopted for a single spiral was slightly changed, to enable fabricating the system in one step, otherwise multiple single spirals would be aligned manually. More in detail, the cutting phase reported in Figure 1 allowed rolling the bi-layered PDMS membrane since we removed the physical constraint on one of its sides, and the silicon wafer at the bottom was already removed in the previous phases. In Figure 4, the cutting process allowed to cut the internal path, thus the three of four edges of the inner spirals, from the bottom silicon wafer, that is still present. Then, in the peeling phase, we rolled the spirals by adding a few drops of water, thanks to the presence of the PVA sacrificial layer, that let them free of constraints with the bottom silicon wafer upon dissolution. There are no phases of light crosslinking in the procedure.

To make the concept clearer, we have added the following sentence in the Materials and Methods, section 4.2. “Fabrication of hierarchical rolled structures”:

“A laser-cut machine was used to cut the three edges of four for the inner spirals on the PDMS bilayer formed onto the PVA-covered silicon wafer, according to Figure 4b. Afterward, a few drops of deionized water were poured on the inner windows of the overall bilayers to let the inner spiral units roll. As the last step, the remaining edge of the bilayer was cut using a scalpel.”

The title of the ms refers to self-foldable but in the test only rolling is described. This is confusing as is the statement in the abstract " radius of spontaneous folding". Although the term "folding" has a number of meanings, the most common of which is "paper folding", where the fold is localised and the folding angle well defined. The introduction has much to say about rolling but nothing about folding . Please alter the title.

We thank the reviewer for this suggestion. According to the reviewer’s comment, we have modified the title in “Modeling self-rollable elastomeric films for building bioinspired hierarchical 3D structures", and removed other incongruent references to self-foldable structures.

Round 2

Reviewer 1 Report

Dear Authors,

Please procced to a final check/polishing of the paper before publication. 

Reviewer 2 Report

The revised version is not suitable for publication